# Improving AlphaFold Predicted Contacts for Alpha-Helical Transmembrane Proteins Using Structural Features

**DOI:** 10.3390/ijms25105247

**Published:** 2024-05-11

**Authors:** Aman Sawhney, Jiefu Li, Li Liao

**Affiliations:** 1Department of Computer and Information Sciences, University of Delaware, Smith Hall, 18 Amstel Avenue, Newark, DE 19716, USA; asawhney@udel.edu; 2School of Optical-Electrical and Computer Engineering, University of Shanghai for Science and Technology, 516 Jun Gong Road, Shanghai 200093, China; lijiefu@usst.edu.cn

**Keywords:** AlphaFold, protein structure, protein structure modeling, Alpha helix, transmembrane proteins, contact map prediction, machine learning, neural networks

## Abstract

Residue contact maps provide a condensed two-dimensional representation of three-dimensional protein structures, serving as a foundational framework in structural modeling but also as an effective tool in their own right in identifying inter-helical binding sites and drawing insights about protein function. Treating contact maps primarily as an intermediate step for 3D structure prediction, contact prediction methods have limited themselves exclusively to sequential features. Now that AlphaFold2 predicts 3D structures with good accuracy in general, we examine (1) how well predicted 3D structures can be directly used for deciding residue contacts, and (2) whether features from 3D structures can be leveraged to further improve residue contact prediction. With a well-known benchmark dataset, we tested predicting inter-helical residue contact based on AlphaFold2’s predicted structures, which gave an 83% average precision, already outperforming a sequential features-based state-of-the-art model. We then developed a procedure to extract features from atomic structure in the neighborhood of a residue pair, hypothesizing that these features will be useful in determining if the residue pair is in contact, provided the structure is decently accurate, such as predicted by AlphaFold2. Training on features generated from experimentally determined structures, we leveraged knowledge from known structures to significantly improve residue contact prediction, when testing using the same set of features but derived using AlphaFold2 structures. Our results demonstrate a remarkable improvement over AlphaFold2, achieving over 91.9% average precision for a held-out subset and over 89.5% average precision in cross-validation experiments.

## 1. Introduction

About 20 to 30 percent of genes in all genomes encode membrane proteins [1,2]. Transmembrane (TM) proteins are involved in essential cell processes such as catalysis, signal transduction, protein targeting and transporting molecules and ions through the cell membrane [3]. In the event of the dysregulation of cellular function, the manipulation of these processes via therapeutic interventions can restore homeostasis [4]. It is therefore no surprise that 60% of all clinically approved drugs target membrane proteins [4].

Understanding the 3D structure of TM proteins is crucial for comprehending their functionality and facilitating the development of drugs [5]. TM proteins are largely α-helical [6]. Generally, there has been a significant sequence-structure gap, and this gap is particularly pronounced when it comes to TM proteins [7]. Since the extraction of membrane proteins from their native lipid environment can alter their integrity and their hydrophobic nature resists water dissolution, preventing crystallization is essential for techniques like X-ray crystallography [5,8]. Though there have been several advances, such as attempts to map the structure while embedding them in a lookalike lipid membrane [9] and making them water-soluble [10], the number of solved structures remains disproportionately low.

When 3D structures are unavailable, a residue contact map provides a simplified 2D representation that is unchanged under translation or rotation and is easily processed by machine learning models. The development of a 3D protein model from contact maps is currently an area of active research. Typically, a folding engine like Rosetta [11] uses binary contact maps as geometric constraints and turns them into folded proteins [12]. In addition, the direct use of residue contact predictions has found applications in enhancing the speed of molecular dynamics simulations [13], in docking simulations [14] and in predicting protein–protein interactions as well [15,16]. TM helices have been observed to tilt and bend when protein structures is captured in different functional states [17]. Hence, a residue contact map could serve as a valuable tool on its own, for detection of inter-helical binding sites, offering insights to proteins’ functions.

In the literature, a range of features derived from physio-chemical attributes, sequence data and co-evolutionary information [7] have been employed to estimate residue contacts. Approaches like EVFold [18] and direct coupling analysis [19], collectively termed as Evolutionary coupling (EC) approaches, compute residue pair co-evolutionary propensities (which correlate with contact propensities) from multiple sequence alignments (MSAs) and have proved more effective than others. Several methods employed supervised learning to combine predictions from various EC methods as input features to improve performance. These include DeepHelicon [7], Wang et al. [20] and DeepMetaPSICOV [21] which use deep learning approaches. Furthermore, studies have indicated that the topological characteristics in the vicinity of a pair of residues within the contact map, including the contact propensities of adjacent positions, can contribute to improving the accuracy of predictions [22] even further.

The use of residual networks (ResNets) [23] with convolutional neural networks (CNNs) greatly improved the quality of the predicted contact maps [12]. Raptor X [20], AlphaFold [24] and TrRosetta [25] all used ResNets for residue contact prediction with great success. An updated RaptorX system [26] predicted discretized inter-residue distances (0.5 Å increments) instead of binary contacts. AlphaFold [24] employed a similar technique, and added components to convert predicted distribution over distances into smooth energy potentials that could be minimized using gradient descent and folded into a 3D structure without the use of a folding engine [12]. These methods take MSAs as direct input and estimate the 3D coordinates of residues using deep learning, thereby delivering an increasingly efficient end-to-end solution.

AlphaFold2 [27], with the use of transformers [28] and sufficiently deep MSAs, has recently demonstrated the capability to achieve near-angstrom accuracy [12]. Given the great success of AlphaFold2, it is conceivable to question whether any other efforts in structural prediction including contact map prediction have become superfluous. Several studies have examined AlphaFold2’s predicted structures, for example to assess the impact of conformational diversity on its predictions [29] or to evaluate if AlphaFold2 learned the physics of folding [30]. In particular, TmAlphaFold [31] examined if AlphaFold2’s predicted alpha-helical TM structures are realistic. They found the quality for a majority of cases (out of 215,844 TM proteins) to be excellent (45.16%) or good (21.51%), and for a lower proportion of proteins, the quality to be fair (25.08%) or poor (2.21%). AlphaFold2 self reports an all-atom accuracy of 1.5 Å r.m.s.d._95_ (95% confidence interval = 1.2–1.6 Å) [27].

Despite AlphaFold2’s high accuracy, there is room for improvement, especially for TM proteins. MULTICOM3 [32] is built on top of AlphaFold2 and AlphaFold-Multimer [33]. It improved upon AlphaFold2’s performance by sampling more structural models via the adjustment of input MSAs and incorporating protein complexes. CGAN-Cmap [34] used a generative adversarial neural network embedded with a series of modified squeeze and excitation residual networks to predict residue contact maps on CASP datasets and achieved a performance gain over contact maps extracted from AlphaFold2.

When it comes to contact map prediction, our previous work shows that information from existing 3D structures could be leveraged to improve prediction accuracy [35]. A classifier trained on structural features extracted from a residue pair’s neighborhood was found to significantly outperform state-of-the-art models using non-structural features, achieving above 90% precision for top L/2 and L inter-helical contacts. In particular, those structural features were also found to be robust to high levels of noise, pessimistically reliable up to 2Å of coordinate noise [35]. It is then intriguing to explore the possibility of applying this idea of using structural features for contact prediction to proteins that do not have an experimentally determined structure but only have decently approximated structures predicted by a computational tool such as AlphaFold2. Here, we explore this idea expanding on our previous work. While AlphaFold2 is not designed for contact map prediction per se, but rather for tertiary structure as a whole, its predicted structure nonetheless can be used to establish a contact map as a by-product. And, therefore, we hypothesize that a general-purpose tertiary structure prediction tool like AlphaFold2 can be “bootstrapped” with features extracted from its predicted structure to perform better for some special purpose tasks such as contact map prediction.

In this work, our aim is to further the utilization of structural features to improve AlphaFold2’s performance for contact map prediction, although AlphaFold2’s performance is typically measured for 3D structures, in terms of predicted local distance difference test (pLDDT) [36]. As previously explained, contact maps are useful on their own; hence, we first evaluate how well AlphaFold2’s predicted structure can deliver for contact point prediction. We found it to be already quite accurate, achieving over 83% average precision for the held-out datasets. We then trained a neural network based classifier on structural features derived from experimentally determined structures and applied the trained classifier to predict residue contacts for proteins with derived features from AlphaFold2’s predicted structures. The results from our experiments show that this method achieved over 91.9% average precision for the held-out datasets, significantly outperforming AlphaFold2 predictions. This pipeline is pictorially depicted in Figure 1. Furthermore, we compared our structure-derived features (SDFs) using 3D coordinates directly (CFs), and found that the latter approach fails to improve upon AlphaFold2 predictions.

## 2. Results

To test out our hypothesis that structure-derived features (SDFs) can help improve the residue contact prediction over AlphaFold2 structure, we designed cross-validation (cv) experiments to train a neural network classifier on residue pairs (contact pairs as positive and non-contact pairs as negative) represented by different types of features—including SDFs and 3D coordinates (CFs)—and evaluated the trained classifier’s performance on both the cv test set and a holdout set. The classification performance is evaluated with two commonly adopted metrics: Average Precision [37] and AUC-ROC [37,38]. For training, the SDFs are derived from the ground truth structures as reported in PDB; whereas for testing, the SDFs are derived from the AlphaFold2-predicted structures, with the intention to make the classifier useful for proteins that do not have ground truth structures. Because the train and test data are from different sources, we also conducted variance analysis based on the statistics of these data to better understand the impact on learning and contact prediction. In addition, a case study of a specific protein is presented with details at residue level to shed light on how the SDF-trained classifier outperforms AlphaFold2 in contact prediction.

### 2.1. Contact Prediction

In Table 1, we report the average 5-fold cross validation performance (repeated 5 times) for contact prediction from the classifier trained with using two feature types—3D coordinates as features (CFs) and structure-derived features (SDFs). For comparison, we also include the performance from AlphaFold2 binary annotations and DeepHelicon predictions. The performance of either feature type, when constructed using experimentally determined structures, is considered as an upper bound. Upper bound performance using our SDFs substantially exceeds using coordinates directly (CFs) by 11.22%, 13.89% and 12.59% for the $S_{L}$, $S_{M2}$ and $S_{M1}$ datasets in terms of average precision; and 0.72%, 0.71% and 0.87% in terms of AUC-ROC. For reproducibility, we report the random seeds used to create the splits in 5-fold cross validation experiments in Appendix A.

**Table 1 ijms-25-05247-t001:** Classification performance—average over 5-fold cross validation (repeated 5 times).

Classifier	Structure Source	Feature Type	$S_{L}$		$S_{M1}$		$S_{M2}$	
			**Average** **Precision**	**AUC-ROC**	**Average** **Precision**	**AUC-ROC**	**Average** **Precision**	**AUC-ROC**
NN (upperbound)	Exp.	SDF	0.9569 ± 0.0039	0.9980 ± 0.0004	0.9497 ± 0.0054	0.9981 ± 0.0004	0.9456 ± 0.0043	0.9983 ± 0.0002
NN	AF	SDF	0.8956 ± 0.0171	0.9919 ± 0.0035	0.9111 ± 0.0204	0.9957 ± 0.0016	0.9038 ± 0.0270	0.9965 ± 0.0011
NN (upperbound)	Exp.	CF	0.8447 ± 0.0193	0.9908 ± 0.0018	0.8238 ± 0.0341	0.9894 ± 0.0041	0.8067 ± 0.4712	0.9912 ± 0.0035
NN	AF	CF	0.8125 ± 0.0246	0.9846 ± 0.0046	0.8349 ± 0.0287	0.9915 ± 0.0017	0.8254 ± 0.0295	0.9927 ± 0.0014
AlphaFold2	-	-	0.7920	0.9441	0.8316	0.9561	0.8473	0.9643
DeepHelicon	-	-	-	-	0.5679 ± 0.0440	0.9337 ± 0.0183	0.5678 ± 0.0479	0.9365 ± 0.0170

Exp—experimentally derived structures; AF—AlphaFold2-predicted structures; SDFs—structurally derived
features; CFs—coordinates as features; NN—neural network architecture presented in Figure 2.

SDFs constructed using AlphaFold2-predicted structures (SDF + AF) outperform AlphaFold2 binary annotations by 10.36%, 5.65% and 7.95% for the $S_{L}$, $S_{M2}$ and $S_{M1}$ datasets, respectively, in terms of average precision; 4.78%, 4.72% and 3.96% respectively in terms of AUC-ROC. Further, SDF + AF comfortably outperforms DeepHelicon by 33.6%, 34.32% for $S_{M2}$ and $S_{M1}$ datasets, respectively, in terms of average precision; 6.00% and 6.20%, respectively, in terms of AUC-ROC.

Prediction results for the held-out datasets ($S_{M2}$ and $S_{M1}$) for both feature types—CFs and SDFs—AlphaFold2 binary annotations and DeepHelicon predictions in Table 2. The upper bound performance using our SDFs substantially exceeds that using coordinates directly (CFs) by 15.51% and 14.77% for the $S_{M2}$ and $S_{M1}$ datasets, respectively, in terms of average precision; 0.68% and 0.79%, respectively, in terms of AUC-ROC.

**Table 2 ijms-25-05247-t002:** Classification performance—held-out datasets.

Classifier	Structure Source	Feature Type	$S_{M1}$		$S_{M2}$	
			**Average** **Precision**	**AUC-ROC**	**Average** **Precision**	**AUC-ROC**
NN (upperbound)	Exp.	SDF	0.9641	0.9986	0.9618	0.9988
NN	AF	SDF	0.9267	0.9958	0.9197	0.9968
NN (upperbound)	Exp.	CF	0.8164	0.9907	0.8067	0.9920
NN	AF	CF	0.7710	0.9891	0.7686	0.9904
AlphaFold2	-	-	0.8316	0.9561	0.8473	0.9643
DeepHelicon	-	-	0.5678	0.9336	0.5678	0.9366

Exp—experimentally derived structures; AF—AlphaFold2-predicted structures; SDFs—structurally derived
features; CFs—coordinates as features; NN—neural network architecture presented in Figure 2.

SDF + AF outperforms AlphaFold2 annotations by 7.25% and 9.5% for the $S_{M2}$ and $S_{M1}$ datasets, respectively, in terms of average precision; 3.25% and 3.96%, respectively, in terms of AUC-ROC. SDF + AF comfortably outperforms DeepHelicon as well by 35.19% and 35.89% for the $S_{M2}$ and $S_{M1}$ datasets, respectively, in terms of average precision; 6.02% and 6.22%, respectively, in terms of AUC-ROC. Further, in nearly all sequences, 98% of the $S_{M1}$ and 97.1% of the $S_{M2}$ datasets (Table 3), the classification performance is improved (measured in terms of average precision). In all experiments, SDFs outperform the baseline CFs.

We report the performance comparison for SDF and CF, in terms of precision and recall at L thresholds, for cross validation experiments—in Appendix A and for held out datasets ($S_{M_{1}}\;\&\;S_{M_{2}}$) in Appendix A. Further, we report per sequence results for the held out datasets ($S_{M_{1}}\;\&\;S_{M_{2}}$) in Appendix A.

Recognizing that the datasets used in this study contain structures with varied resolutions, which may impact how our proposed method works, we repeated the experiments on datasets with stratified analysis: structures with high resolution (≤2.5Å) and structures with low resolution (2.5Å to 3.5Å). The results (detailed in Appendix A) are quite comparable and consistent improvements are seen in each case, with a slight variation: the high resolution set had a higher baseline (AlphaFold2) and relatively smaller improvement (5 to 6 percentage points) whereas the low resolution set had a lower baseline and relatively larger improvement (8 to 9 percentage points). We report the fraction of the structures with high resolution and low resolution in each dataset ($S_{L},\;S_{M_{1}}\&S_{M_{2}}$) in Appendix A. We also provide the random seeds used for cross validation in this experiment in Appendix A.

### 2.2. Variance Analysis

The classifier is trained on features constructed from experimentally derived structures. However during testing, only features constructed from AlphaFold2-predicted structures will be available to us. Consequently, the classifier’s testing performance depends on whether the feature distributions from the two data sources (experimental vs. AlphaFold2 prediction) are similar. In Table 4, we report the feature mean—average across all features and samples—and feature variance—standard deviation across all features and samples—for the $S_{L}$, $S_{M1}$ and $S_{M2}$ datasets. The datasets were first scaled to a range of [−1,1]. It can be seen that SDFs constructed using structures predicted by AlphaFold2 or experimentally determined structures are very similar, differing by 0.013, 0.017, 0.035 for the $S_{L}$, $S_{M1}$ and $S_{M2}$ datasets, respectively, in terms of feature mean; −0.005, 0.003, −0.005, respectively, in terms of feature variance. CFs constructed using AlphaFold2 structures or experimentally determined structures vary more, differing by 0.207, 0.380 and 0.014 for the $S_{L}$, $S_{M1}$ and $S_{M2}$ datasets, respectively, in terms of feature mean; 0.142, 0.067 and 0.049, respectively, in terms of feature variance. These statistics are helpful in gauging the distribution similarity. Using relative residue distance and angles thus potentially has the effect of scaling for mean removal and variance scaling. In other words, it is likely that an efficient model would need to predict relative angles and distances that are closer in distribution to experimental determined ones; hence, SDFs are a natural fit. CFs exhibit higher variance when generated using experimentally determined structures, which makes intuitive sense as one would expect real residue coordinates to exhibit more variance than predicted ones.

We further examine this divergence (defined in Appendix A) of two data sources via a second auxiliary classifier’s ability to differentiate between features generated using the two sources (AlphaFold2 and experimental) in Appendix A [40,41,42,43,44,45]. The results reported in Appendix A support what the simple statistics (means and variance) have revealed.

The contact prediction performance for held-out datasets ($S_{M1}$ and $S_{M2}$) is higher than corresponding cross validation experiments. We attribute this to a bigger training set size. Performance comparison for individual sequences, recall and precision scores [46,47,48] at the top L, L/2, L/5, L/10 thresholds (top k residue pair predictions set as 1 s and the rest as 0 s; L represents the combined sequential length of transmembrane helices within a sequence) are reported in Appendix A.

DeepHelicon dataset consists of structures that were experimentally determined prior to the release of AlphaFold DB; it is likely they were part of AlphaFold’s training, which then bolsters our case.

### 2.3. Case Study

Additionally, in Appendix A, we illustrate, via a case study of the chain 4g7vS [49] from dataset $S_{L}$, how using a classifier trained on SDFs from experimentally derived features can improve AlphaFold’s predicted structure.

## 3. Discussion

In this study, we adopted an unorthodox approach of extracting features in the neighborhood of a residue pair from experimentally determined structures and used them to train a classifier for predicting contacts between residues located on different helices of α-helical TM proteins. This approach, which is in contrast to most other works that have focused on developing methods to predict residue contact based on the primary structure, would not be useful should the atomic structures be not available. What we demonstrated here is that AlphaFold2 has dramatically raised the quality of predicted structures—in our held-out experiments, we found it to be highly accurate, achieving over 83% average precision—and can be used as a surrogate of ground truth 3D structure for providing informative structural features. We trained on features generated from experimentally determined structures and predicted on features constructed using AlphaFold2-predicted structures. The results from our experiments demonstrate a significant improvement over AlphaFold2, achieving over 91.9% average precision for both $S_{M1}$ and $S_{M2}$ datasets. Based on what is demonstrated in this study, it is conceivable that more sophisticated structural features may be extracted from AlphaFold2 structures to potentially lead to further improvement. It is worth noting that we also show that simply training on coordinates directly does not lead to a performance improvement. Structurally derived features potentially reduce distributional distance between features derived from experimentally determined and predicted structures. This work demonstrates that a residue sequence neighborhood is information-rich, can be used to produce more accurate structures and that features derived from a residue’s structural neighborhood can be generalized across sequences. As a future work, it is possible that we may leverage the improved contact map to enhance the predicted structures even further.

## 4. Materials and Methods

### 4.1. Dataset—Experimentally Determined Structures

We adopted the widely used DeepHelicon dataset [7] for this study. It was created with TM protein chains from the PDBTM database [50], each of the selected 5606 α-helical chains had a resolution finer than 3.5Å. Further, the chains were non-redundantly curated using a 23% sequence identity threshold and with a maximum TM score [51] of 0.4 to ascertain that the protein chains were structurally dissimilar. The resulting dataset consists of 222 protein chains, featuring a differing count of TM helices (2–17). It is segmented into three sub-datasets: (a) TRAIN—165 sequences that serve as the training set, which we refer to as dataset $S_{L}$ for clarity; (b) TEST—57 sequences that serve as a held-out set, which we refer to as dataset $S_{M1}$ for clarity; and (c) PREVIOUS—44 sequences that serve as an additional held-out set, which we refer to as dataset $S_{M2}$ for clarity [52,53]. For every protein chain, annotations indicating which residue pairs are in contact and which positions are within the TM region, protein sequence, and the 3D structure in PDB format, which includes the atomic coordinates of each residue’s heavy atoms, are included with the dataset. Additionally, DeepHelicon’s model predictions for the held-out datasets (TEST and PREVIOUS) are included.

Given a chain’s atomic structure, a residue pair is considered to be in contact if their heavy atoms are within a specific distance of each other. In the DeepHelicon dataset [7], a contact point is defined as 2 residues that are separated by a minimum of 5 residues in sequence and for which the minimum distance between any pair of their heavy atoms measures less than 5.5Å [7].

Following our previous work [35], a few sequences are removed—those are sequences with no inter-helical contact points or with positions annotated to be in TM zone not matching positions used by DeepHelicon (refer to the Appendix A)—this results in 162, 40 and 57 sequences in $S_{L}$, $S_{M2}$ and $S_{M1}$ datasets, respectively. A summary of these changes can be found in Table 5.

### 4.2. Dataset—AlphaFold Predicted Structures

AlphaFold DB provides predicted structures for over 200 million protein sequences in the UniProt [54] reference proteome [36,55]. These structures can be accessed via the protein chain’s UniProtKB ID [54], and include atomic coordinates of each residue’s heavy atoms in PDB format. We relied on the Research Collaboratory for Structural Bioinformatics protein data bank (RCSB PDB (RCSB.org)) [56,57] to map the PDB ID of every chain in the DeepHelicon dataset to UniProtKB ID. If a match was found, the corresponding predicted structure was accessed via AlphaFold DB. For several protein chains, an integer offset to PDB positions in the DeepHelicon dataset is needed to sequentially align them with AlphaFold structures [58] (refer to Appendix A). In case a UniProtKB ID match was not found in RCSB PDB or the sequences from UniProt and DeepHelicon dataset matched partially, i.e., all positions annotated to be in TM zones were not contiguously included, then the chain was removed from the dataset (refer to the Appendix A). This process resulted in 154, 34 and 49 sequences in the $S_{L}$, $S_{M2}$ and $S_{M1}$ datasets, respectively.

These modifications, as well as the contact ratio (CR) (for residue pairs situated on distinct TM helices and separated by at least 5 residues in the sequence), are presented in Table 5. The definition of CR is provided in Equation (Equation 1).
(1)$$CR=\frac{\#contact\;points}{\#residue\;pair\;positions}$$

As mentioned in Section 4.1, the DeepHelicon dataset includes annotations indicating residue positions located in the TM zone. For matching structures obtained from AlphaFold DB, we adopt the same annotations. Following the contact definition described in Section 4.1, for matching predicted structures obtained from AlphaFold DB, we generated annotations indicating which residue pair positions are contact points.

### 4.3. Methods

The methods proposed for predicting residue contact maps consist of mainly two parts: selecting features and training a classifier. In the following, we show in detail how to construct a feature vector from a 3D structure, which is either experimentally determined or computationally predicted, to represent a residue pair, and how to use them to feed into a neural network-based classifier for training.

#### 4.3.1. Structurally Derived Features (SDFs)

Following our previous work [35], we employ structural features derived from coordinate data for predicting residue contacts. For inter-helical contact, only residue pairs (i,j) (*i* and *j* represent positions in the amino acid sequence) that are on different helices and separated by a minimum of 5 residues are considered, which is the criterion from [7] where we obtained the data. To predict whether (i,j) is a contact, we gather features from its neighborhood, which comprises of 8 positions in a window of size 3×3 centered at (i,j): (i,j±1), (i±1,j), (i±1,j±1); specifically, for each neighboring position in this window, a vector of 5 features is constructed, including the relative residue distances, relative residue angle and inter-helical tilt angle. And we concatenate features for these eight neighbors to create a feature vector of size 40 (resulting from 8 positions each with 5 features). Features from (i,j)) are excluded so that the classifier does not rely on the distance between residue *i* and residue *j* to determine contact, as this distance is how a residue pair is named as being a contact or not. This process is illustrated in Figure 3a. More detailed descriptions of these extracted features are provided in the following subsections.

#### Inter-Helical Tilt Angle (θ)

The inter-helical tilt angle for a pair of residues is the angle measured between the helices on which these residues are located [59]. Within an α-helix, each spiral turn of the backbone coil takes about 4 residues. All C=O groups are oriented in one direction while all N−H groups are oriented in the opposite direction; thus, the dipoles are consistently aligned. The planes of the peptide bonds are approximately parallel with the helical axis and, at the same time, amino side chains project outwards from the central helical axis typically oriented towards the amino-terminal end [60]. Motivated by this observation, we determine the orientation of a helical axis by calculating the average direction of the vector C(i)=O(i)−N(i+4) for all residues within the helix. The inter-helical tilt angle is the angle that describes the orientation difference between the axes of two helices, and hence can be very informative regarding how two helices may interact with each other. We use the Pymol package for these computations [61,62,63]. A diagrammatic representation is provided in Appendix A.

#### Relative Residue Distance

We detail three features that describe the relative distance between residues:$D_{1}$ distance (mean relative residue distance) [35,64,65]: We calculate the average Euclidean distance between a pair of residues by considering all paired combinations of their heavy atoms. If $\left\{A_{x}^{1},\cdots A_{x}^{M}\right\}$ are the 3D coordinates of the residue $R_{x}$ and $\left\{A_{y}^{1},\cdots A_{y}^{N}\right\}$ for the residue $R_{y}$. Additionally, if dist(i,j) represents the Euclidean distance between two sets of 3D coordinates *i* and *j* then the mean relative residue distance between a residue pair $\left(R_{x},R_{y}\right)$ is
(2)$$D_{1}\left(R_{x},R_{y}\right)=\;\frac{1}{MN}\sum_{i=1}^{M}\sum_{j=1}^{N}dist\left(A_{x}^{i},A_{y}^{j}\right)$$$D_{1}$ deviation (relative residue distance deviation) [35,64,65]: We consider the distances between all paired combinations of a residue pair’s heavy atoms and calculate the standard deviation for these distances. Then, deviation of the relative residue distances between a residue pair $\left(R_{x},R_{y}\right)$ is
(3)$$SD_{D_{1}}\left(R_{x},R_{y}\right)=\sqrt{\left\{\frac{1}{MN}\sum_{i=1}^{M}\sum_{j=1}^{N}\left[{\left(dist\left(A_{x}^{i},A_{y}^{j}\right)-D_{1}\left(R_{x},R_{y}\right)\right)}^{2}\right]\right\}}$$$D_{\alpha }$ (Relative $C_{\alpha }$ distance) [35,64,65]: We calculate the Euclidean distance between the alpha carbons of a pair of residues. If the $k^{th}$ atom for a residue *R* is returned by a function atom(R,k). Additionally, if $C_{\alpha }$ is the $i^{th}$ atom for residue $R_{x}$, i.e., $atom\left(R_{x},i\right)=C_{\alpha }$ and the $j^{th}$ atom for residue $R_{y}$ i.e. $atom\left(R_{y},j\right)=C_{\alpha }$. Then, relative $C_{\alpha }$ distance between a residue pair $\left(R_{x},R_{y}\right)$ is
(4)$$D_{\alpha }\left(R_{x},R_{y}\right)=\;dist\left(A_{x}^{i},A_{y}^{j}\right)$$

#### Relative Residue Angle (δ)

We define a residue’s plane using vectors formed by the $C_{\alpha }$ to *N* atom and the $C_{\alpha }$ to *C* atom in the carboxyl group [65]. For a pair of residues, the relative residue angle is defined as the absolute angle between the surface normals of their respective planes [35]. A diagrammatic representation is provided in Appendix A.

It is important to note that the definition of a residue pair being a contact point relies on the minimum distance between paired combinations of their heavy atoms. However, during the prediction process, we utilize structural information from the neighborhood of the residue pair, and employ different distance functions ($D_{1}$ distance and $D_{\alpha }$ distance) to determine if it is a contact point.

#### 4.3.2. Coordinates as Features (CFs)

To demonstrate the effectiveness of our derived features described above, we also directly use 3D coordinates of residue pair’s heavy atoms as features. This serves as a performance baseline. Residue pairs (i,j) ) (*i* and *j* represent positions in the amino acid sequence) that fulfill the criteria of being sequence separated by a minimum of 5 residues and present on different helices (inter-helical) are the only ones considered. For each of the eight positions in the neighborhood window of size 3 centered at (i,j) (not including (i,j)), a vector consisting of x, y, z coordinates of the heavy atoms from the residue pair of interest (size 24) is constructed. Each residue is represented by the x,y,z coordinates of 4 heavy atoms from its structure, namely nitrogen atom (*N*) from the amino group, alpha carbon ($C_{\alpha }$), oxygen from the carboxyl group (*O*) and beta carbon ($C_{\beta }$). We concatenate features for these eight neighbors to create a feature vector of size 192 (resulting from 8 positions each with 24 features). This process is illustrated in Figure 3b.

#### 4.3.3. Classification Experiment

We handled the prediction of an inter-helical TM residue pair position being a contact point as a binary classification problem using supervised learning. As mentioned earlier, we only consider residue pair positions that fulfill the criteria of being sequence separated by a minimum of 5 residues and present on different helices (inter-helical). For structurally derived features, we constructed a feature vector of length 40 (described in Section 4.3.1). While using coordinates as features, a feature vector of length 192 was formed (described in Section 4.3.2).

Features from either feature set (structurally derived or coordinates) were first normalized to a [−1,1] scale before being used for classification, such that $f_{i_{scaled}}^{t}=-1+2\times \left(\frac{f_{i}^{t}-min\left(f_{i}\right)}{max\left(f_{i}\right)-min\left(f_{i}\right)}\right)$ where the $t^{th}$ sample for the feature $f_{i}$ is denoted by $f_{i}^{t}$, and the functions min(.) and max(.) determine the lowest and highest observed value for the feature $f_{i}$. Additionally, for the feature $f_{i}$, $t^{th}$ sample’s scaled value is represented by $f_{i_{scaled}}^{t}$.

We constructed a neural network classifier consisting of 6 hidden layers with leaky Relu activation function [66] to capture the non-linearity in the features and used binary cross entropy as the loss criterion at the output. The architecture is depicted in Figure 2. Using Adam optimizer [67] with a learning rate of 0.0001, we trained in batches of 256 samples for a total of 400 epochs. The weights of the network were initialized using Xavier uniform distribution [68] and gradients were clipped to the range [−1,1] to prevent exploding and vanishing gradients [69]. We used the PyTorch package for our implementation [70].

A static fully connected linear layer was used to project structurally derived features from 40 to 192 dimensions; this enabled us to use the same network for both (structural derived and coordinates) feature sets.

We assessed our performance on each dataset—$S_{L}$ (154 sequences), $S_{M1}$ (49 sequences) and $S_{M2}$ (34 sequences) using cross validation (5 folds) [71,72]. In each fold, 20% of the sequences were randomly selected and set aside for validation, while the remaining 80% were used for training. Further, a model was retrained on the entire $S_{L}$ dataset and its performance evaluated on the held-out $S_{M2}$ and $S_{M1}$ datasets.

In each experiment, we used features (SDFs or CFs) constructed from experimentally determined structures during training and, for comparison purposes, tested the trained classifier on two separate cases: (a) features constructed from experimental determined structures, and (b) features constructed from AlphaFold-predicted structures.

#### Performance Metrics

We evaluated the classification performance with the following two widely used metrics:Average precision: Average precision condenses the precision–recall curve by taking a weighted average of precision values at various thresholds. The weight applied to each threshold’s precision value is determined by the increase in recall from the previous threshold [37].
(5)$$Average\;Precision=\sum_{n}\left(R_{n}-R_{n-1}\right)P_{n}$$
where precision at the *n*th threshold is denoted by $P_{n}$ and recall by $R_{n}$. For predicted structures from AlphaFold DB, we generate binary annotations for whether a residue pair is a contact point (described in Section 4.2). In Equation (Equation 5), this is the case when there is only one (n=1) threshold and, AveragePrecision=P×R; where *P* and *R* are the observed precision and recall scores using these binary labels.AUC-ROC: The area under the receiver operating characteristic curve is calculated using the trapezoidal rule [37,38].

These two metrics allow us to evaluate a classifier’s predictive power without imposing a threshold on the prediction score so that an overall assessment can be achieved, not tied to a specific threshold choice. Once the test examples are ranked by their prediction score from a classifier, an ROC curve can be plotted the true positive rate as a function of false positive rate by running down the ranked list as follows: (a) at each position in the list, predict the test examples above as positive and below as negative, (b) compare the prediction with the ground truth label to determine true positive and false positive, and (c) calculate the rates and move to the next position in the list. The higher the curve—more true positives predicted at a given false positive rate—the better the performance, which is measured as the area under the curve, a value (called ROC score) between 0 and 1, with 1 being the perfect performance and 0.5 being a performance comparable to a random toss-up. Using a similar procedure running down the ranked list of test examples, a curve can be plotted with precision as a function of recall. Average precision is essentially the area under the precision–recall curve. It has been reported [73] that for skewed data with a much larger proportion of negative examples, which is our case, ROC scores tend to be more optimistic than the actual performance is and, in instances like this, average precision may present a more realistic picture. For both metrics, we provide the average score across all sequences.

## Figures and Tables

**Figure 1 ijms-25-05247-f001:**
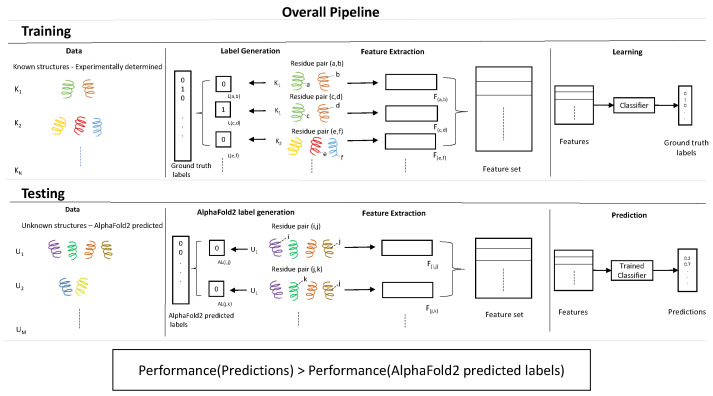
Graphical depiction of the overall pipeline.

**Figure 2 ijms-25-05247-f002:**
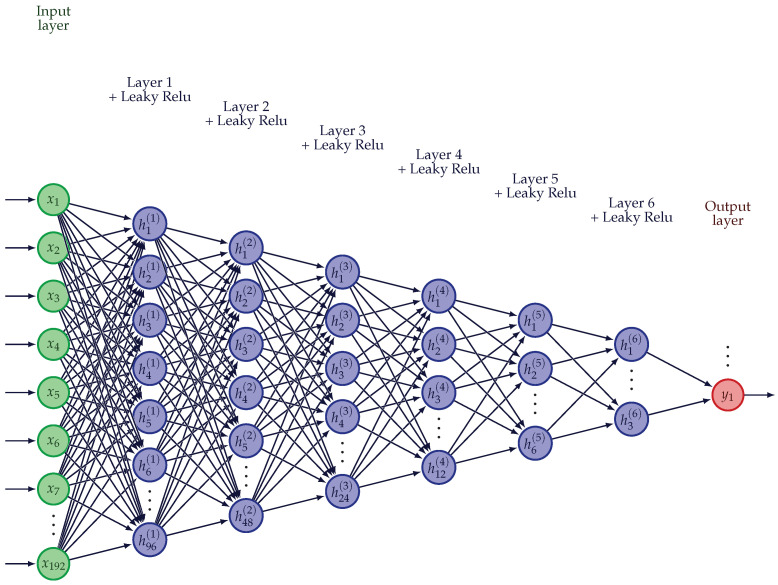
Neural network architecture (adapted from [39]).

**Figure 3 ijms-25-05247-f003:**
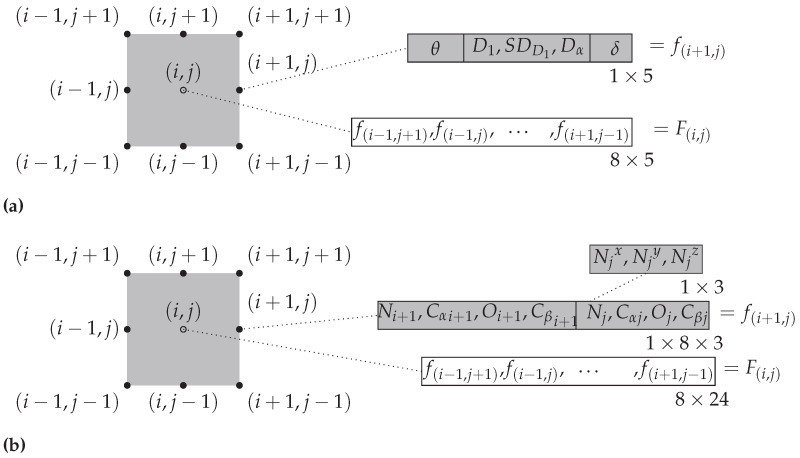
Feature vectors are features derived from a 3×3 window of residue pairs surrounding and centered on a specific residue pair (i,j)(not including (i,j)). (First published in [35]). (**a**) Structurally derived features—feature vector of length 40. (**b**) Coordinates as features—feature vector of length 192.

**Table 3 ijms-25-05247-t003:** Individual sequences improved (in terms of average precision)—held-out datasets.

Dataset	# Seqs (% of Total)
$S_{M1}$ (49)	48 (98.0)
$S_{M2}$ (34)	33 (97.1)

**Table 4 ijms-25-05247-t004:** Feature mean and variance of AlphaFold2-predicted and experimental structures.

		$S_{L}$		$S_{M1}$		$S_{M2}$	
**Structure** **Source**	**Features**	**Feature** **Mean**	**Feature** **Variance**	**Feature** **Mean**	**Feature** **Variance**	**Feature** **Mean**	**Feature** **Variance**
Exp	SDF	−0.1616	0.3656	−0.2487	0.3363	−0.1634	0.3653
AF	SDF	−0.1744	0.3707	−0.2655	0.3338	−0.1979	0.3707
Exp	CF	−0.1716	0.3025	−0.2293	0.3466	0.0275	0.2969
AF	CF	0.0351	0.1604	0.1510	0.2799	0.0132	0.2474

Exp—experimentally derived structures; AF—AlphaFold2-predicted structures; SDF—structurally derived
feature; CFs—coordinates as features; NN—neural network architecture presented in Figure 2.

**Table 5 ijms-25-05247-t005:** Dataset statistics—protein chain count and contact ratio for $S_{L}$, $S_{M1}$ and $S_{M2}$ datasets.

Dataset	#Sequences	#Filtered Sequences	AF Available	CR×100
$S_{L}$	165	162	154	2.10
$S_{M1}$	57	54	49	2.07
$S_{M2}$	44	40	34	1.95

AF available—a matching AlphaFold2-predicted structure was found; CR—contact ratio.

## Data Availability

Alphafold2 structures used in this work and code are available at https://www.eecis.udel.edu/~lliao/helical_contact (accessed on 10 May 2024).

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
