# Peer review of "Improving AlphaFold Predicted Contacts for Alpha-Helical Transmembrane Proteins Using Structural Features"

_ijms, 2024, doi:10.3390/ijms25105247_

Round 1
Reviewer 1 Report
Comments and Suggestions for Authors
This paper presents an approach for predicting inter-residue contact maps of alpha-helical transmembrane proteins. This could prove to be a useful addition to existing tools for 3D structure prediction of transmembrane proteins. The authors show how the use of a classifier trained on the structural features derived from experimental data can improve the structures predicted by AlphaFold. The paper is well written and the results are presented in an understandable way, so I recommend the publication in IJMS after a revision.
In introducing relative distances between residues, the authors omit to choose measures such as the minimum inter-atomic distance between residues, which seems to more naturally represent a contact between two residues. The authors should at least discuss and comment on the possibility of using other measures to describe structural features.
Reviewer 2 Report
Comments and Suggestions for Authors
The manuscript submitted by Aman Sawhney and colleagues describe a computational tool for predicting contact in alpha-helical transmembrane proteins. It improves the AlphaFold2 performance by including a series of stereochemical variables that describe the local environment of residues pairs. In my opinion, this manuscript is interesting and should be made available to the scientific community. However, several modifications are needed.
Major issues
(a) I suggest using the expression “3D structure” instead of “3d structure” or “3-d structure”. In addition, I suggest to avoid the use of “configuration” and to prefer “structure”.
(b) Structures at low resolution, for example worse than 2.5 Angstroms, should be discarded, since the accuracy of the atomic positions is very low, especially for the side-chains.
(c) It is unclear why two data sets (SM1 and SM2) have been used instead of only one.
(d) The terms in equation 1 should be better explained and the term “domain” in line 268 is inappropriate.
(e) The description of the Structurally derived features should be rewritten in a more fluent way (it is basically impossible to understand without looking at Figure 2a.
(f) Line 297: H-bonds in helices do not stabilize helices. Once the helix is unfolded, they are broken. However, new H-bonds, of similar energy, are formed with water molecules.
Minor issues
(g) Line 43: delete “to estimate”.
(h) Lines 44-46: the sentence Hence, a residue contact map … are compared [17]” seems to be unnecessary.
(i) Line 47: the expression “to a protein’s functions” might become “proteins’ functions”.
(j) Line 126: “Average Precision Average” might become “Average Precision”.
(k) A general enhancement in the quality of the English is recommended.
Comments on the Quality of English Language
A general enhancement in the quality of the English is recommended.
Round 2
Reviewer 2 Report
Comments and Suggestions for Authors
The revised version of this manuscript is acceptable for publication.